# miR-203 secreted in extracellular vesicles mediates the communication between neural crest and placode cells required for trigeminal ganglia formation

Yanel E. Bernardi[1,2], Estefania Sanchez-Vasquez[1,2], Rocío Belén Márquez[1,2], Michael L. Piacentino[3], Hugo Urrutia[3], Izadora Rossi[4], Karina L. Alcântara Saraiva[5], Antonio Pereira-Neves[6], Marcel I. Ramirez[4], Marianne E. Bronner[3], Natalia de Miguel[2,7], Pablo H. Strobl-Mazzulla[1,2] *

1 Laboratory of Developmental Biology, Instituto Tecnológico de Chascomús (INTECH), CONICET-UNSAM, Chascomús, Argentina, 2 Escuela de Bio y Nanotecnologías (UNSAM), Chascomús, Argentina, 3 Division of Biology, California Institute of Technology, Pasadena, California, United States of America, 4 Laboratorio de biologia molecular e sistematica de tripanossomatideos, Instituto Carlos Chagas, Fiocruz Parana, Curitiba, Brazil, 5 Núcleo de Plataformas Tecnológicas, Instituto Aggeu Magalhães, Fiocruz, Recife, Brazil, 6 Departamento de Microbiologia, Instituto Aggeu Magalhães, Fiocruz, Recife, Brazil, 7 Laboratorio de Parásitos Anaerobios, Instituto Tecnológico Chascomús (INTECH), CONICET-UNSAM, Chascomús, Argentina

* strobl@intech.gov.ar

## Abstract

While interactions between neural crest and placode cells are critical for the proper formation of the trigeminal ganglion, the mechanisms underlying this process remain largely uncharacterized. Here, by using chick embryos, we show that the microRNA (miR)-203, whose epigenetic repression is required for neural crest migration, is reactivated in coalescing and condensing trigeminal ganglion cells. Overexpression of miR-203 induces ectopic coalescence of neural crest cells and increases ganglion size. By employing cell-specific electroporations for either miR-203 sponging or genomic editing using CRISPR/Cas9, we elucidated that neural crest cells serve as the source, while placode cells serve as the site of action for miR-203 in trigeminal ganglion condensation. Demonstrating intercellular communication, overexpression of miR-203 in the neural crest in vitro or in vivo represses an miR-responsive sensor in placode cells. Moreover, neural crest-secreted extracellular vesicles (EVs), visualized using pHluorin-CD63 vector, become incorporated into the cytoplasm of placode cells. Finally, RT-PCR analysis shows that small EVs isolated from condensing trigeminal ganglia are selectively loaded with miR-203. Together, our findings reveal a critical role in vivo for neural crest-placode communication mediated by sEVs and their selective microRNA cargo for proper trigeminal ganglion formation.

**Data Availability Statement:** All relevant data are within the paper and its Supporting Information files.

**Funding:** This work was supported by the Agencia Nacional de Promoción Científica y Tecnológica (PICT 2018-1879 to P.H.S-M.) and by the Fogarty International Center of the National Institutes of Health (R21TW011224 to M.E.B. and P.H.S-M.). The funders had no role in study design, data collection and analysis, decision to publish, or preparation of the manuscript.

**Competing interests:** The authors have declared that no competing interests exist.

**Abbreviations:** EMT, epithelial-to-mesenchymal transition; EV, extracellular vesicle; HH, Hamburger and Hamilton; IHC, immunohistochemistry; ISH, in situ hybridization; MET, mesenchymal-to-epithelial transition; MVB, multivesicular body; NC, neural crest; NT, neural tube; NTA, nanoparticle tracking analysis; PM, plasma membrane; sEV, small extracellular vesicle; TEM, transmission electron microscopy; TG, trigeminal ganglion.

## Introduction

Organogenesis requires the coordinated interaction of different cell types. In vertebrates, a good example is the interaction between neural crest (NC) cells and ectodermal placodes, 2 cell types of distinct embryonic origin that both contribute to the formation of cranial ganglia such as the trigeminal ganglion (TG). The TG is the largest ganglion in the head and is responsible for mediating sensation of pain, touch, and temperature in the face as well as innervating the sensory apparatus of the eye muscles and the upper and lower jaws [1,2].

The NC, a population of multipotent cells specified in the dorsal neural tube (NT) of vertebrates embryos, migrates extensively and differentiates into diverse cell types including neurons and glia of the peripheral nervous system [3]. Placode cells arise from the surface ectoderm [4,5], then ingress or invaginate into the cranial mesenchyme. Placode cells then interact with NC cells during condensation of the cranial ganglia, producing functional ganglia comprised of both neural crest- and placode-derived cells [6–8]. While both NC and placode cells contribute to trigeminal neurons, NC cells are the sole source of peripheral glia [7,9].

NC and placode cells are known to interact extensively; for example, *Xenopus* NC cells chase placodal cells by Sdf-mediated chemotaxis, and placodal cells are repulsed by a PCP and N-cadherin signaling mechanism [10,11]. Thus, precise cell–cell communication is required to facilitate mixing, proper positioning, aggregation, and differentiation of the forming cranial ganglia [6]. However, surprisingly little is known about the nature of interactions between NC and placode cells during ganglion formation.

In recent years, extracellular vesicles (EVs) have emerged as a novel mode of cell-to-cell communication. EVs are capable of transferring information from a donor cell to a recipient cell, leading to changes in gene expression and cell function [12–17]. EVs are classified in 3 main types based on their size and biogenesis: exosomes (50 to 150 nm in diameter) that arise from multivesicular bodies (MVBs), ectosomes or shedding vesicles derived from the plasma membrane (PM) of cells through direct outward budding (100 to 1,000 nm in diameter), and apoptotic bodies that are released during apoptotic cell death (bigger than 1,000 nm in size, [18]). In particular, the subset of small EVs (sEVs) have been extensively studied in cancer cells, where exosomes and other EVs have been shown to drive multiple aspects of cancer metastasis, including the promotion of cancer cell motility, invasiveness, and premetastatic niche seeding [19–21]. sEVs contain proteins, RNAs, and the cargo of specific microRNAs (miRNAs) that can be transferred from a donor to a recipient cell, leading to changes in gene expression and cellular function in the receivers [12–17].

miRNAs are a class of small noncoding RNAs that regulate gene expression at the posttranscriptional level mostly by binding to the 3′ UTR of target transcripts and fine-tune their expression through degradation and/or translational repression [22,23]. They are known to play important roles in the fine regulation of gene expression during normal development, from gastrulation to complex organ formation. miRNAs specifically regulate epithelial plasticity by promoting both epithelial cell delamination and mesenchymal cell coalescence [24]. In particular, we have previously shown that miR-203 is epigenetically repressed by DNA methylation in premigratory NC cells, thus allowing the up-regulation of its target genes, *Snail2* and *Phf12*, which are essential for their epithelial-to-mesenchymal transition (EMT) [25,26]. Importantly, we observed that the miR-203 locus is rapidly demethylated during NC migration. At the end of migration, NC cells condense into different derivatives, like the cranial sensory ganglia. Thus, we hypothesized that ganglion condensation may require the reactivation of miR-203. Using the trigeminal ganglion as a model, we show that miR-203 is re-expressed in coalescing and condensed NC cells to regulate trigeminal ganglion assembly. Intriguingly, we find that miR-203 is required for trigeminal ganglion condensation. Further, we find that

miR-203-containing sEVs are produced in NC cells, which are then taken up by placode cells in which the miRNA exerts its biological effect. Together, our findings reveal that miR-203 is up-regulated in post-migratory neural crest cells and its transport into placode cells via small EVs is critical for trigeminal ganglion condensation.

## Materials and methods

### Embryos

Fertilized chicken eggs obtained from "Escuela de Educación Secundaria Agraria de Chascomús" were incubated at 38˚C until the desired embryonic stage according to the criteria of Hamburger and Hamilton (HH) [27].

### DNA construct and electroporation

For in ovo electroporation, eggs were incubated horizontally until stage HH8/9 and vectors were injected and electroporated to exclusively target the neural crest or the trigeminal placode cells (S1 Fig). After injection on the specific sites, a platinum electrode was placed on each side of the embryo, and the chick embryos were electroporated with 5 pulses of 15 V for 50 ms on and 100 ms off intervals. The electroporated eggs were then sealed with adhesive tape and incubated until the desired stages were reached as described previously [28]. For ex ovo electroporation, embryos at HH4/5 were injected using pressure system but platinum electrodes were placed vertically across the chick embryos and electroporated with 5 pulses of 5.5 V. Embryos were cultured in 0.5 ml albumen in tissue-culture dishes until the desired stages as previously described [29]. Embryos were then removed from eggs or dishes, placed in PBS and fixed in 4% PFA and utilized for immunohistochemistry or in situ hybridization protocol. The miR-203 overexpressing, sponge and sensor vectors were described and characterized in our previous publication [25]. We employed the optimized CRISPR/Cas9 system for chick embryos developed by Gandhi and colleagues [30] to knockout miR-203. Two gRNAs (gRNA_101: CGCTGGTCAATGGTCCTAAACATTTCAC; gRNA_87: GCCCGGGCCTC GCTGGTCAAG) were designed to target the miR-203 locus. These gRNAs were expressed under the control of the optimized chicken U6.3 promoter. Additionally, we designed a scrambled gRNA to serve as a control. To validate CRISPR-mediated genomic editing at the miR-203 locus and assess the reduction in its expression, we co-electroporated vectors containing the gRNAs (scrambled gRNA on the left side and both miR-203 gRNAs on the right side) with CAGG>nls-Cas9-nls at the gastrula stage. Embryos were then harvested at stage HH8 (S1B Fig) and dissected in halves. Small RNAs were isolated for stem-loop RT-qPCR, and genomic DNA was prepared for genotyping. The PCR-amplified miR-203 locus flanking the gRNA target sites (Fw: CCCCAGCGCGAGGACGTT; Rv: CAGCCCTCGATTCGCGCACT) was subjected to heteroduplex mobility assay (HMA PCR). Electrophoresis was performed on a 12% acrylamide gel, followed by staining with ethidium bromide for 15 min. Multiple heteroduplex bands were observed in PCR amplicons from miR-203 gRNA-treated embryos, and a single band obtained in scrambled gRNA-electroporated embryos. Consistent with these findings, a significant reduction in miR-203 expression was evident in embryos treated with miR-203 gRNAs compared to those treated with scrambled gRNA. pHluo_M153R-CD63-mScarlet (a gift from Alissa M. Weaver, Vanderbilt University School of Medicine, Nashville, Tennessee, United States of America, [31]) was amplified with 2 pairs of primers (pHluo-Fw: 5′-AAA ctc gag GCC ACC ATG GCG GTG GAA GGA G-3′; pHluo-Rev: 5′-AAA gct agc CTA GGA TCC CTT GTA CAG CTC GTC C-3′), digested (XhoI/NheI), and subcloned into the chick overexpressing pCIG vector. The pCIG-mRFP, containing a CAG promoter and a membrane RFP, was utilized to target placode cells in explants co-culture experiments.

## In situ hybridization (ISH)

Embryos were fixed overnight in 4% PFA in PBS at 4°C and then utilized for whole-mount ISH for mRNA [32] and miRNAs [25] following the previously published protocols. In both cases, the mRNAs and LNA probes were labeled with digoxigenin (Roche). Hybridized probes were detected using an alkaline phosphatase-conjugated anti-digoxigenin antibody (Roche, 1:2,000) in the presence of NBT/BCIP substrate (Roche). After ISH, some embryos were re-fixed in 4% PFA in PBS, washed, embedded in gelatin, and cryostat sectioned at a thickness of 14 to 16 μm. Embryos were photographed as a whole-mount using a ZEISS SteREO Discovery V20 Stereomicroscope with an Axiocam 512 camera and Carl ZEISS ZEN2 (blue edition) software.

## Immunohistochemistry (IHC)

Whole embryos were fixed for 15 min in 4% PFA, washed in TBST (500 mM Tris-HCl (pH 7.4), 1.5 M NaCl, 10 mM $CaCl_2$, and 0.5% Triton X-100) and blocked in 5% FBS in TBST for 1 h at RT. Embryos were then incubated in mouse anti-Tuj1 (1:250; Covance) and/or mouse anti-HNK1 (1:10; supplied by the Developmental Studies Hybridoma Bank) overnight at 4°C diluted in TBST-FBS. Secondary antibodies were goat anti-mouse IgG2a Alexa Fluor 647 (1:500) and goat anti-mouse IgM Alexa Fluor 568 (1:500) for 1 h at room temperature. After several washes in TBS-T, whole embryos or sections (provided by Lic. Gabriela Carina López from the "Histotechnical Service" at INTECH) were mounted and imaged by using Carl ZEISS Axio observer 7 inverted microscope containing an Axiocam 503 camera and Carl ZEISS ZEN2.3 (blue edition) software.

## Extracellular vesicles isolation and characterization

Trigeminal ganglia were dissected and collected from approximately 80 to 100 HH17-19 stage embryos for each of the replicates required for the different characterization techniques. For TEM, a group of trigeminal ganglia was fixed in a 4% (v/v) glutaraldehyde solution in 0.1 M (v/v) cacodylate buffer, pH 7.2 to 7.4. Samples were gradually dehydrated with serial solutions of 50%, 70%, 80%, 90%, 95%, 100% acetone and then embedded in epoxyPolybed 8120 resin. Next, ultra-thin sections (approximately 70-nm thick) were harvested on 300 mesh copper grids, stained with 5% uranyl acetate and 1% lead citrate, and observed with a FEI Tecnai G2 Spirit transmission electron microscope, operating at 120 kV. The images were randomly acquired with a CCD camera system (MegaView G2, Olympus, Germany).

For preparation of sEVs, trigeminal ganglia were treated with a solution of dispase and trypsin to obtain a single cell suspension. Samples were centrifuged at $10,000 \times g$ for 10 min and the supernatant was recovered to isolate sEVs. Then, the sample was filtered through 0.2 μm filter and then pelleted by centrifugation at $100,000 \times g$ for 90 min to obtain an sEVs enriched fraction. As the protocol of EVs isolation include a filtration step using a 0.2 μm filter, the term sEVs (that include exosome and small size microvesicles) will be used throughout the text. sEVs were resuspended in PBS-DEPC containing a protease inhibitor cocktail (cOmplete ULTRA Tablets, Mini, EASYpack, Sigma). For characterization of particle quality size and abundance of the isolated sEVs nanoparticle tracking analysis methodology (NTA—Nanoparticle Tracking Analysis, Nanosight LM10 (Malvern, United Kingdom)) was used. The samples were read in triplicate for 60 s at 10 frames per second, analyzed using NTA Software (version 2.3). The results were annotated as concentration (particles/ml) and mode size (nm). In addition, a fraction of the sample was reserved for the isolation of small RNA.

## RNA preparation and RT-PCR

RNA was prepared from the trigeminal ganglion and sEV fraction using the RNAqueous-Micro isolation kit (Ambion) following the manufacturer's instructions, and RNA was treated with amplification-grade DNaseI (Invitrogen). The reverse transcription reaction to obtain the cDNA was performed with the MystiCq microRNA cDNA Synthesis Mix kit (Merck) and amplified by PCR using the following primers (miR-34-5p Fw: 5′-GCC GCT GGC AGT GTC TTA G-3′; miR-203 Fw: 5′-CCG GCG TGA AAT GTT TAG G-3′; and miR-UNI Rev: 5′-GAG GTA TTC GCA CCA GAG GA-3′). On the other side, to assess for the impact of the miR-203 sponge on miR-203 expression, we employed the stem-loop-RT-qPCR method, as detailed in our previous work [25]. In brief, embryos hemi-electroporated ex ovo with miR-203 and Scramble sponges (S2 Fig) were incubated until HH8, followed by the dissection of dorsal NTs. Total RNA extraction was accomplished using the RNAqueous-micro kit (Ambion), and the reverse transcription of specific miRNAs (miR-203 and miR16 utilized as normalization control miRNA) was carried out using SuperScript II (Invitrogen) with stem-loop-miRNA-specific primers (slo-miR-203: 5′-GTCTCCTCTGGTGC AGGGTCCGAGGTATTCGCACCAGAGGAGACCAAGTG-3′; slo-miR-16: 5′-GTCT CCTCTGGTGCAGGGTCCGAGGTATTCGCACCAGAGGAGACCAAGTG-3′). Subsequent quantitative PCRs were executed on a 96-well plate qPCR machine (StepOne), employing SYBR green with ROX (Roche).

## Explants co-culture

Neural crest and placode cells were electroporated separately in ovo at stage HH9. After treatment, the embryos were allowed to grow at 37˚C and each tissue was dissected. A neural crest explant was placed next to a placodal explant in plates previously treated with fibronectin and containing DMEM medium supplemented with 10% fetal bovine serum and penicillin/streptomycin. The explant pairs were cultured at 37˚C in 5% $CO_2$ overnight. Six co-culture experiments were imaged for 2 h with a Zeiss LSM 980 inverted confocal microscope at 37˚C in 5% $CO_2$.

## Quantification of trigeminal ganglia areas

Images were transformed to binary scale and the Fiji area calculation function was utilized to measure the area as marked by *Sox10* ISH. The background was considered and subtracted from the whole-mount images of stages HH17-18 heads. Areas of the ganglia in the treated sides were normalized to the control side for each embryo as described [33].

## Results

### miR-203 is expressed in coalescing trigeminal ganglion cells

We first examined the expression pattern of miR-203 during the course of cranial NC migration, coalescence and condensation during trigeminal gangliogenesis (see scheme in Fig 1A) by performing in situ hybridization (ISH) analysis at selected developmental stages in chick embryos. miR-203 was previously shown to be present in premigratory NC cells but down-regulated prior to their delamination from the NT [25]. In agreement with this, we noted that mature miR-203, detected using locked nucleic acid-digoxigenin-labeled probes, was absent at HH13 from migrating HNK1 immunoreactive cranial NC cells (Fig 1B and 1B'). However, miR-203 expression was again noticeable at stage HH16 (Fig 1C and 1C') co-localizing with NC (red arrowheads HNK1+) and placode cells (green arrowheads Tuj1+) when both populations are coalescing at the site of trigeminal ganglion formation. Later at stage HH20 when the

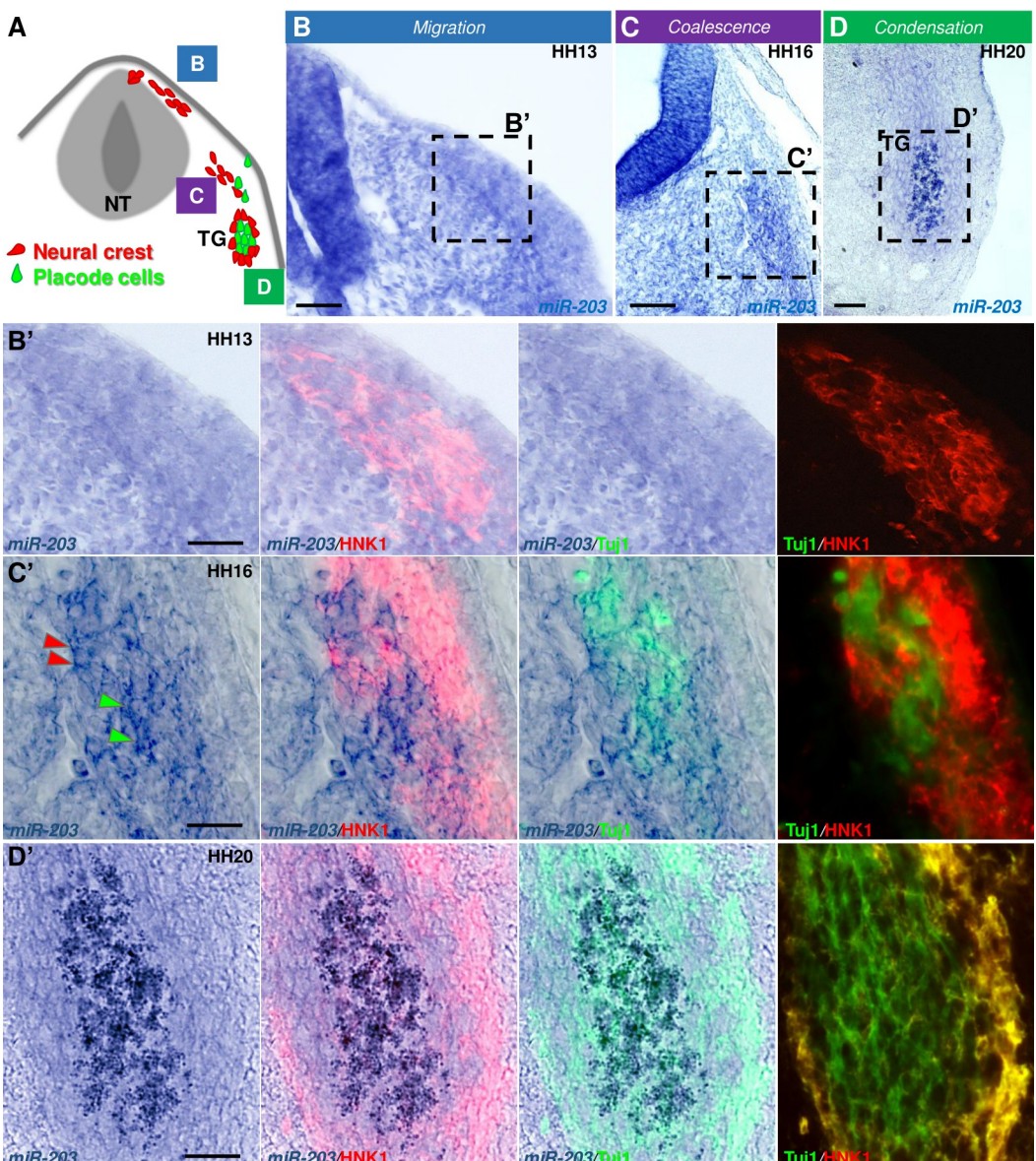

**Fig 1. miR-203 expression is activated during trigeminal ganglion formation.** (A) Schematic diagrams of a cross-section through the cranial NT illustrating the migration (B, B') of NC and their coalescence (C, C') and condensation (D, D') with placode cells during TG formation in chicken embryos. Scale bars: 100 μm. Transverse section of miR-203 after in situ hybridization using an LNA-DIG-labeled probe followed by immunostaining for HNK1 (neural crest marker in red) and Tuj1 (placodal marker in green) at HH13 **(B)**, HH16 **(C)**, and HH20 **(D)**. Scale bars: 50 μm. While miR-203 is absent from migrating NC cells, its expression reinitiates at the time of ganglion coalescence and remains present in the condensed ganglion. Red and green arrowheads in C' denoted early coalescing NC and placode cells expressing miR-203, respectively. NC, neural crest; NT, neural tube; TG, trigeminal ganglion.

ganglion is almost fully condensed, signal was robust, particularly at the center of the lobe where most of the placode cells are residing (Fig 1D and 1D'). Taken together, these data indicate that miR-203 expression is reactivated at the time of NC coalescence and condensation into ganglia, consistent with the intriguing possibility that miR-203 may be required for NC aggregation during trigeminal ganglion formation.

## Overexpression of miR-203 generates ectopic aggregation of NC cells and a more condensed trigeminal ganglion

Given that miR-203 expression is down-regulated at the onset of NC migration and then re-expressed during trigeminal ganglion formation, we asked whether overexpression of miR-203 would accelerate the condensation process. To this end, we generated an overexpression vector containing pre-miR-203 [25] that was electroporated into the right half of the NT at HH9. Embryos were then allowed to develop until HH15-16. The results show that excess miR-203 causes ectopic aggregation of NC cells, as identified by in situ hybridization against *Sox10*, compared with the control side or embryos treated with empty vector (Control OE) (Fig 2A,

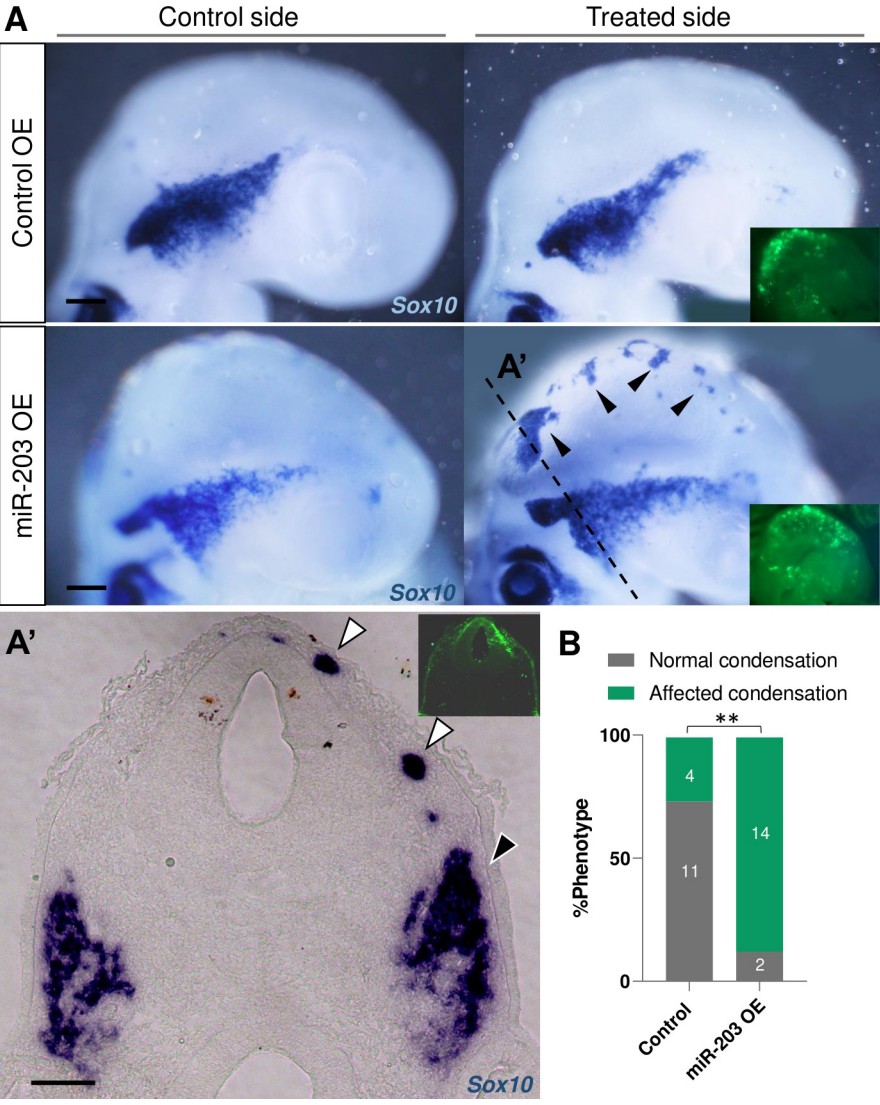

**Fig 2. Overexpression of miR-203 promotes ectopic and premature NC condensation and enhanced aggregation into the trigeminal ganglion. (A)** In situ hybridization for *Sox10* in HH15-16 embryos unilaterally electroporated (treated side showing green fluorescence in insets) with miR-203 (miR-203 OE) or empty control (Control OE) overexpression vectors. **(A')** Cross section of miR-203 OE embryo at the level shown in A (dotted line) reveals an ectopic Sox10+ group of cells (white arrowhead) and a denser TG (black arrowhead) on the treated side. Scale bars: 100 μm. **(B)** Quantification of embryos showing a phenotype (normal versus affected condensation having ectopic condensation and/or denser TG) on Control and miR-203 OE. Numbers in the graph represent the numbers of analyzed embryos. **P = 0.001 by contingency table followed by a $\chi^2$ test. TG, trigeminal ganglion.

black arrowheads). In transverse section, ectopically aggregation of NC cells (Fig 2A', white arrowheads) and a more densely packed trigeminal ganglion (Fig 2A', black arrowhead) are evident on the treated side compared to control side (Fig 2B). These results suggest a possible role for miR-203 during trigeminal ganglion condensation.

## NC is the source of miR-203 transcription, but it places of action are the placode cells

Given that miR-203 re-initiates during trigeminal condensation, we next asked whether its loss of function would disrupt proper ganglion formation. To test this possibility, we utilized a "sponge" vector containing repeated miR-203 antisense sequences (miR-203 sponge) previously validated [25] to sequester endogenous miR-203, thereby diminishing its expression (S1 Fig). A sponge vector containing an miR-203 scrambled sequence (Scrambled sponge) was utilized as a control. The NT was electroporated at stages HH8 to target NC cells, but not placodal cells, in one side of the embryo (S2A Fig), and then examined after the ganglia had condensed (HH17-18). To measure the level of condensation, we quantitated the trigeminal ganglia area after ISH for *Sox10* in treated versus control side of each embryo. The analysis was conducted using whole-mount images at stages HH17-18. Surprisingly, we failed to detect significant differences in the area of the trigeminal ganglion after the loss of miR-203 in NC cells (Fig 3A).

The trigeminal ganglion has a dual origin from both NC and ectodermal placodal cells. Therefore, we next explored the possible functional role of miR-203 in the trigeminal placodes, but not affecting NC cells, by electroporating the right placodal ectoderm at HH9 (S2B Fig) with the miR-203 or scrambled sponge plasmids. Intriguingly, the miR-203 sponge resulted in trigeminal ganglia that displayed a more loosely organized and less aggregated morphology than those in the non-injected side or observed in control embryos (Fig 3B, black arrowhead).

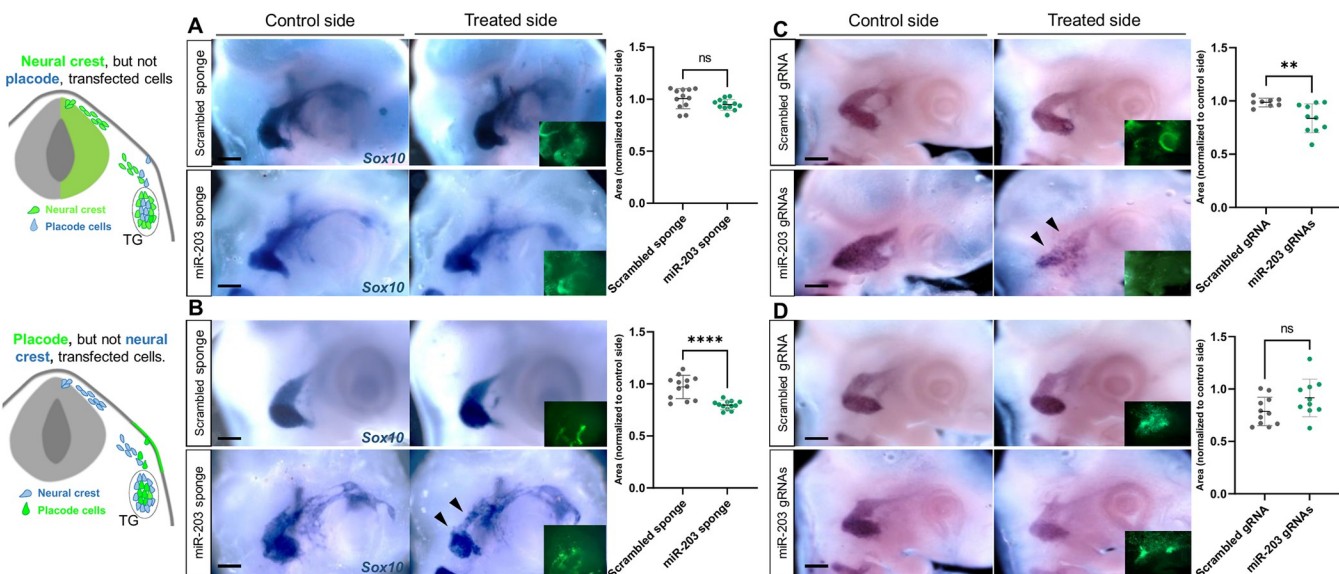

**Fig 3. miR-203 is produced in NC but it places of action are placode cells to secure normal trigeminal ganglion condensation.** Electroporation schemes for region-specific transfections (see also S2 Fig). Embryos at HH9 were unilaterally electroporated (treated side showing green fluorescence in insets) in the NT or ectodermal placodes with miR-203 sponge (*n* = 12) or scrambled sponge (*n* = 12) plasmids **(A, B)**; or with miR-203 gRNAs + Cas9 (*n* = 10) or Scrambled gRNAs + Cas9 (NT *n* = 8; ectodermal placode *n* = 11) **(C, D)** (see also S1 Fig and S1 Data). Trigeminal ganglia condensation was analyzed by ISH for *Sox10* at HH17-18. Scale bars: 100 μm. Scatter plots represent the quantification of trigeminal ganglia areas (treated side versus control side). ****$P$ < 0.0001, **$P$ = 0.0085 by two-tailed unpaired *t* test. ns, non-significant differences. Values are means ± SD. NC, neural crest; NT, neural tube.

This result is consistent with a significant reduction in the ganglia-occupied area in embryos treated with the miR-203 sponge compared to those treated with scrambled sponge. Our findings raise the intriguing possibility that miR-203 is produced in the neural crest (donor cell) but exerts its biological effect in the placode cells (recipient cell). To demonstrate this, we now employed CRISPR/Cas9 system to genomic editing the miR-203 locus and reduce its expression (S1B Fig). Interestingly, the electroporation of miR-203 gRNAs or scrambled gRNAs in placode cells resulted in normal trigeminal ganglia formation (Fig 3D). However, the electroporation of miR-203 gRNAs in NC resulted in a less aggregated morphology resulting in a significant reduction in the ganglia-occupied area compared to the non-electroporated side of the same embryos and with those treated with scrambled gRNA (Fig 3C, black arrowhead). Taking together, these results are consistent with the putative role of placode cells as crucial mediators of NC condensation [34], thus emphasizing the importance of cellular communication between the NC and placode cells to ensure correct aggregation in time and space to form the trigeminal ganglion.

## Extracellular vesicles produced by NC cells are taken up by placode cells

Small EVs, including exosomes, have been described as a novel mode of cell-to-cell communication [13,20]. In particular, sEVs can transport subsets of miRNAs, among other molecules, from a donor cell to a recipient cell, modifying gene expression and cell function in the latter [35–37]. With this in mind, we evaluated the production of sEVs during trigeminal ganglion condensation as a potential mediator of NC-placode communication. To this end, we first performed transmission electron microscopy (TEM) from dissected trigeminal ganglia at HH17. The results reveal cells in close contact within the condensing ganglion (Fig 4A). These contain MVBs with intraluminal vesicles located in the cytosol close to the PM (Fig 4B and 4C). Some of MVB appear to be releasing exosomes-like structures into the extracellular space (Fig 4B). Shedding vesicles protruding from the PM are also observed (Fig 4C).

As a further confirmation that EVs produced by the NC cells are able to reach and be internalized by placode cells, we have adapted the pHluo_M153R-CD63-mScarlet vector [31] to work in chick embryos. This plasmid allows dynamic subcellular monitoring of exosome life-cycle, including MVB trafficking and exosome uptake. The plasmid contains a modified pH-sensitive GFP sequence (pHluo) inserted into the first extracellular loop of the tetraspanin CD63. Of note, pHLuo-CD63 does not fluoresce in the acidic endosomal pH of MBV; however, once exocytosed into the neutral pH of the extracellular environment, it emits a bright fluorescence. In addition, the plasmid is tagged with pH-insensitive red fluorescent protein (mScarlet) that allows the visualization of exosome trafficking (see scheme in Fig 4D).

pHluo-CD63-mScarlet was introduced into the NC by electroporating into the NT of HH9⁻ chick embryos. After NC migration and TG condensation, we observed active vesicle release in the TG condensing region (dashed white line bordering the ganglia) compared with the midbrain region where only mScarlet fluorescence was detected (Fig 4E, white arrowheads). Transverses sections through these embryos reveal mesenchymal migratory NC cells (HNK1 +) containing MVBs (S3A and S3A' Fig). Particularly, we observed migratory NC cells having MVBs in acidic conditions (mScarlet only) and in neutral pH during the secretion process (mScarlet and pHluo positive) (Fig 4F). Similar to what has been previously described [31], we observed trails that may represent EVs deposition. Finally, to demonstrate the ability of NC cells to release EVs that can reach placode cells, sections of electroporated embryos with pHLuo-CD63-mScarlet were immunostained at HH16 for the neuronal marker Tuj1, which will mark placodally derived neurons at this stage (Fig 4G). Intermingled neural crest (pHluo +) and placode cells (Tuj1+) were detected in the region of trigeminal condensation, with

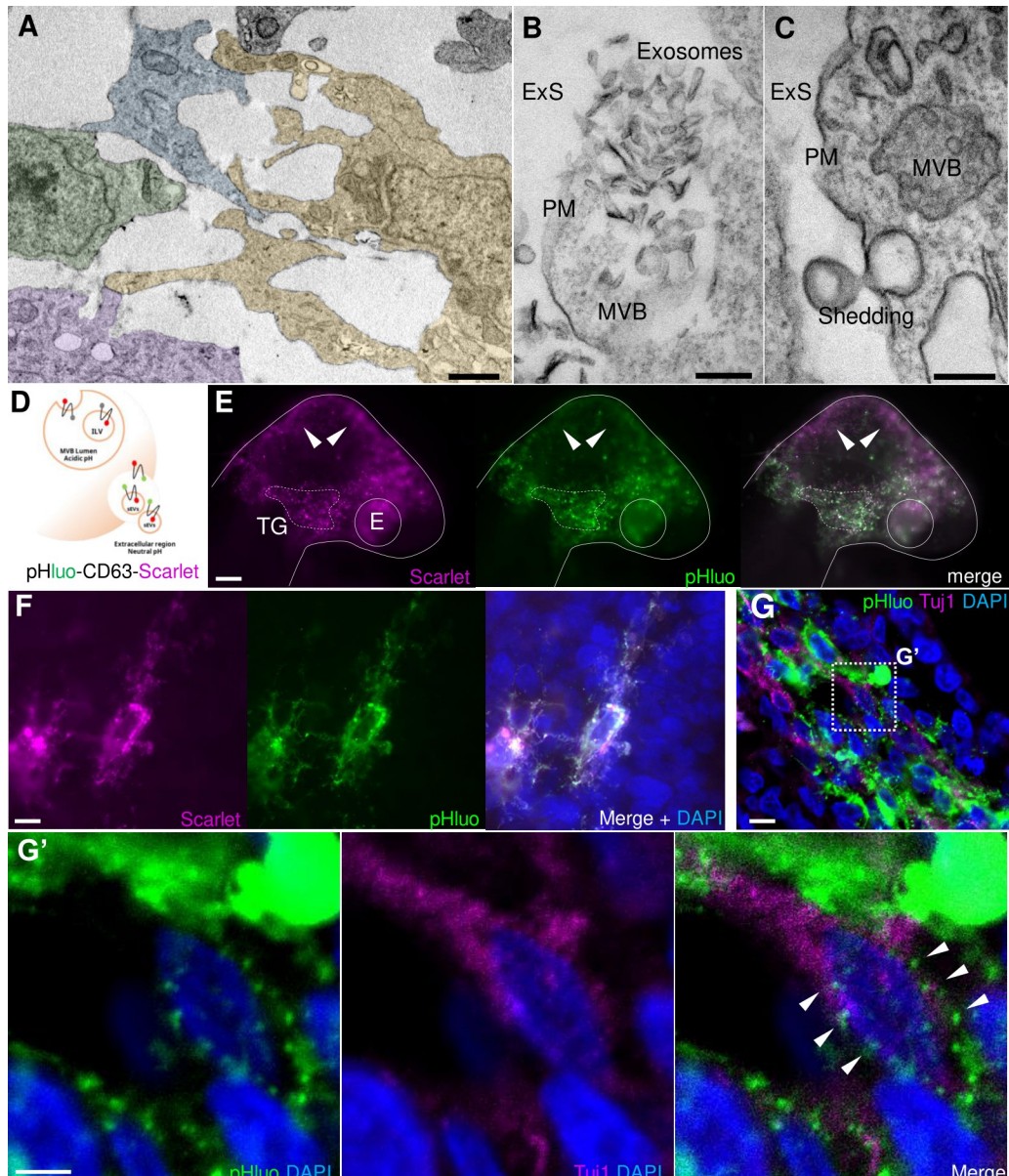

**Fig 4. Extracellular vesicles are actively released by NC cells in the vicinity of placode cells during TG condensation.** TEM from dissected trigeminal ganglia at HH17 shows that cells (identify each with different colors) have many direct cell-to-cell contacts to each other **(A)**, the presence of MVBs releasing exosomes-like structures into the extracellular space (ExS) **(B)**, and a shedding vesicle protruding from PM **(C)**. Bars: 2 μm (A), 200 nm (B and C). **(D)** Diagram of pHluo-CD63-mScarlet functional assay. **(E)** Embryos electroporated in the NT with pHluo-CD63-mScarlet at HH9- and visualized at stage HH16 exhibited GFP+/mScarlet+ fluorescence in the condensing trigeminal ganglion area (dotted line); in contrast, cells located in the midbrain are enrich in mScarlet+ (white arrowheads). Scale bar: 100 μm. TG, trigeminal ganglion; E, eye. **(F)** Migratory neural crest cells electroporated with pHluo-CD63-mScarlet exhibit intraluminal MVB (pHluo-/mScarlet+), extracellular/releasing EVs (pHluo+/mScarlet+) and deposit trails behind the cells. Scale bar: 20 μm. **(G)** Transverse section through the TG showing NC cells electroporated with pHluo-CD63-mScarlet and placode cells immunostained for Tuj1 (magenta). Scale bar: 10 μm. **(G')** Zoom of box in G shows a placode cells (Tuj1+) with pHluo puncta incorporated into their cytoplasm (white arrowheads). Scale bar: 20 μm. EV, extracellular vesicle; MVB, multivesicular body; NC, neural crest; PM, plasma membrane; TEM, transmission electron microscopy.

45 ± 9.6% of the latter containing pHluo+ puncta incorporated into their cytoplasm (Figs 4G', white arrowheads and S3B). Taken together, these results demonstrate cellular interaction mediated by EVs between NC and placode cells in vivo at the time of trigeminal ganglion condensation.

To gain deeper resolution and reveal interactions in real-time, we next performed co-cultures with NC and placode explants from embryos electroporated with pHluo (pseudocolored yellow) and pCIG-mRFP (pseudocolored magenta), respectively (Fig 5A). The explant pairs were cultured until migratory cells from both populations contacted one another at which time we generated time-lapse movies of the co-cultures. Live imaging revealed NC cells that were surrounded by numerous extracellular pHluo+ puncta (possibly EVs deposits), as well as trails (possibly migrasomes or retraction fibers) and cytonemes-like structures (Fig 5B and S1 Movie). Interestingly, the cytoneme-like structures produced by migratory NC cells were in dynamic contact with placode cells (Fig 5B'). In this sense, it has been shown that small EVs can travel along cytonemes and are released in close proximity to recipient cells during development [38,39], raising the intriguing possibility that may be similar during neural crest–placode interactions.

It was previously shown that exosomes could be captured by filopodia or macropinocytosis events and endocytosed by recipient cells [40]. We observed a similar event in which placode cells contained vesicular structures in their membrane (macropinosomes) (Fig 5C, white arrowheads; Z-stack in S2 Movie) and cytoplasm (Fig 5D and 5D', white arrowhead; time lapse and z-stacks in S3 and S4 Movies, respectively) containing pHluo puncta coming from co-cultured NC cells. Interestingly, we also noted that placode cells produced filopodia that move toward EVs deposits and engulf them (Fig 5E and 5F, white arrowhead; time lapse and 3D rotation in S5 and S6 Movies, respectively). These observations demonstrate that neural crest-produced EVs are internalized by placode cells.

## miR-203 produced in NC cells regulates translation in recipient placode cells

To determine whether miR-203 produced in NC cells can reach placode cells to exert a biological effect, we designed an experiment in which we drove overexpression of miR-203 and cytoplasmic EGFP in the NC cells by electroporation into the premigratory NC. In the same embryos, we electroporated the trigeminal placode in the ectoderm with a dual-colored sensor vector. This vector expresses both nuclear d4EGFPn and mRFPn, but the first containing 2 mature miR-203 recognition sites such that the miR-203, but not other miRNA, can bind and affect only d4EGFPn translation (S4 Fig) [25]. Embryos were electroporated at HH9- with the 2 vectors and allowed to grow until HH17 (see scheme in Fig 6A). Transverse sections through the embryos were then immunostained for Tuj1 to identify the trigeminal ganglion cells (Fig 6B). Although some of the placode cells reaching the condensing area were EGFP+/RFP+ (white arrowhead), some cells within close proximity to the NC (black arrowhead) were only RFP+ (dotted circles) (Figs 6B' and S5A). This reduced EGFP expression is not visible in the non-migrating ectodermal cells where all the cells co-express both EGFP and RFP cells (white arrowheads in Fig 6B). Importantly, embryos overexpression miR-203 displayed a significant reduction compared with controls when the EGFP/RFP fluorescence intensity ratios were quantified on placode cells (Fig 6D).

A similar experiment to that shown in Fig 5A was performed where separate embryos were electroporated either with miR-203 or control overexpression vectors in the NT, and the dual-sensor vector was electroporated in the trigeminal placode. Explants from both dorsal NT and ectodermal placode were co-cultured ex vivo until migratory cells from both populations were

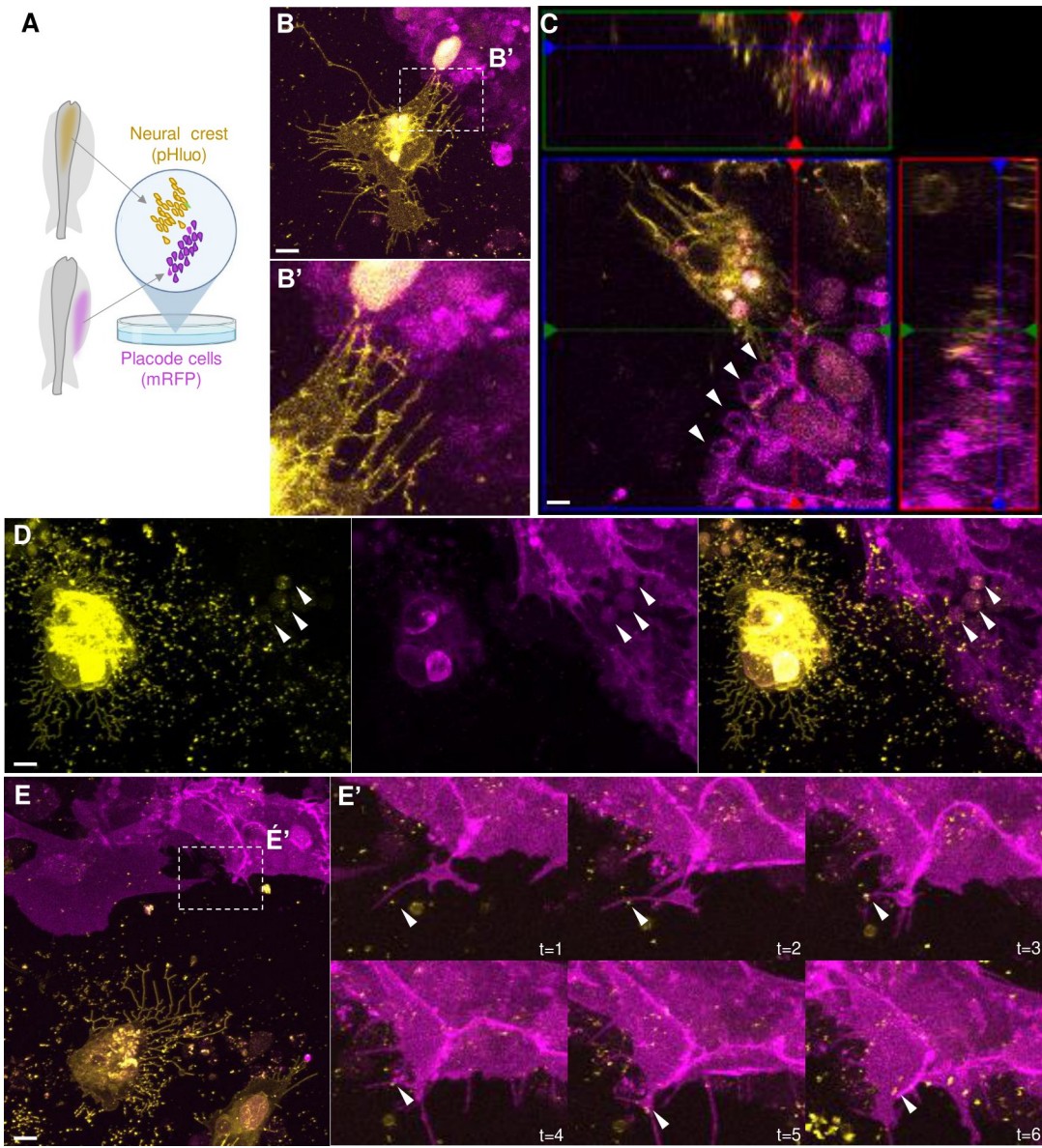

**Fig 5. Extracellular vesicles produced by NC cells are engulfed by placode cells in explant co-cultures. (A)** Scheme of a chicken embryo at HH9⁻ with NT electroporated with a pHluo (pseudocolored in yellow) and placode cells electroporated with membrane RFP (pseudocolored in magenta), respectively. The electroporated dorsal NT (mostly containing NC cells) and ectodermal trigeminal placode cells were dissected at HH9⁺ and HH10⁻, respectively, and co-cultured to examine their interactions. **(B)** NC cells (yellow) exhibiting trails (possibly migrasomes or retraction fibers) and cytonemes-like structures in very close contact with placode cells (magenta). **(B')** Higher magnification of box in B showing the cytonemes from NC cells contacting placode cells. See also **S1 Movie**. **(C)** Placode cells showing vesicular structures like macropinosomes in their membrane (white arrowheads). See also z-stack in **S2 Movie**. **(D, D')** NC cells expressing pHluo were surrounded by numerous extracellular pHluo+ puncta (possibly EVs deposits) and trails (possibly migrasomes or retraction fibers). Co-cultured placodal cells showing intra-cytoplasmic vesicular structures containing pHluo+ puncta (white arrowheads). See also **S3 and S4 Movies**. **(E)** Placode cell co-cultured with NC cells expressing pHluo produce filopodia that move toward the EVs deposits and engulf them. **(E')** Zoom of box in E showing the time-lapse showing placode cells engulfing a pHluo+ puncta (white arrowhead). See also **S5 Movie**. **(F)** 3D reconstruction of placode protrusion observed in (E) demonstrating the internalized pHluo+ puncta (white arrowhead). See also **S6 Movie.** Scale bars: 10 μm. EV, extracellular vesicle; NC, neural crest; NT, neural tube.

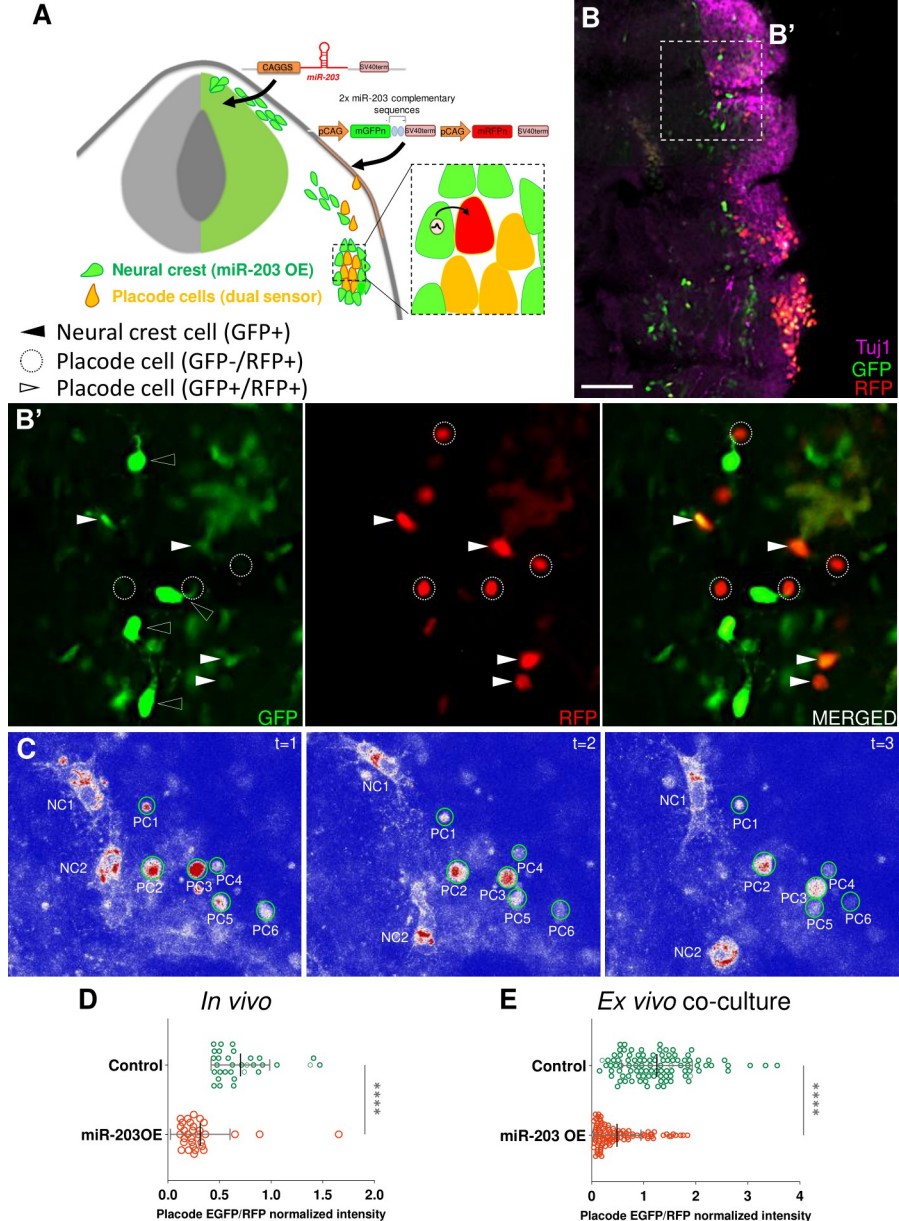

**Fig 6. miR-203 generated in NC cells inhibits a sensor target electroporated in placode cells both in vivo and ex vivo.** **(A)** Scheme of chick embryo with NT electroporated with miR-203 or empty control overexpressing vectors (with cytoplasmic EGFP) and placode cells electroporated with a dual-colored sensor vector (pSdmiR-203) containing 2 copies of complementary sequences to the mature miR-203. pCAG, Chick β-actin promoter; d4EGFPn nuclear-localized destabilized EGFP with a half-life of 4 h; mRFPn, nuclear-localized monomeric red fluorescent protein. Box of the condensing trigeminal ganglion indicates area of enlarged inset exemplifying the transfer of miR-203 from NC (green) to placode cells (orange because of the expression of both EGFP and RFP), thus affecting the EGFP translation (red cell). **(B)** Transverse section of HH17 embryos immunostained for Tuj1 (magenta), to identify the trigeminal ganglia with coalescing NC (electroporated with miR-203 OE vector) and placode (electroporated with the dual-colored sensor vector; see white arrowheads) cells. **(B')** Zoom of box in B identifying migratory NC cells with cytoplasmic EGFP+ (black arrowhead) and placode cells with nuclear EGFP+/RFP+ (white arrowhead) or only RFP+ (dotted circles). Scale bar: 100 μm. See also S5A Fig for DAPI staining. **(C)** Time-lapse of co-cultured placode ectodermal cells (electroporated with the dual-colored sensor vector) and dorsal NT (electroporated with miR-203 OE) explants. Placodal cells (PC1-6) interact with NC cells (NC1-2) where the EGFP channel was pseudocolored (red>white>blue) to visualize the EGFP decay in placode cells over time. **See also S5B Fig for control and S7 Movie**. **(D)** Scatter plot from in vivo experiments analyzing the EGFP/RFP intensity for individual placode cells reaching the condensing area at HH17 from miR-203 OE or Control electroporated embryos (cells counted in 2–5 sections per

embryo, $n$ = 4 embryos from 2 independent electroporations; see S1 Data). ****$P$ < 0.0001 calculated using unpaired Student's $t$ test. Values are means ± SD. (**E**) Scatter plot from ex vivo co-cultured experiment analyzing the EGFP/RFP normalized intensity for individual placode cells interacting with NC cells from miR-203 OE or Control electroporated embryos (100 cells, $n$ = 4 co-cultured explants for each treatment from 2 independent experiments; see S1 Data). ****$P$ < 0.0001 calculated using unpaired Student's $t$ test. Values are means ± SD. NC, neural crest; NT, neural tube.

in contact. LUT pseudocolored cells for EGFP intensity (RED>WHITE>BLUE) were analyzed by time-lapse demonstrating that placode cells (with nuclear EGFP+/RFP+) lose EGFP intensity over time when they are in close proximity to the NC cells overexpressing miR-203 (Fig 6C and S7 Movie). The decrease in EGFP intensity was not observed in control explants, where the NC cells were electroporated with an empty vector (S5B Fig). In addition, EGFP/RFP intensity ratios in placode cells co-cultured with miR-203 overexpressing NC cells were quantified and showed a significant decrease compared with controls (Fig 6E).

Taken together, our in vivo and co-cultured explant results demonstrate that miR-203 produced in the NC reaches and suppresses translation in placode cells, thus supporting the idea that miRNAs may act as intercellular signals mediating proper neural crest–placode communication.

## Endogenous miR-203 is specifically loaded into sEVs produced during TG condensation

To determine the possibility that miR-203 may be loaded into EVs released in the context of trigeminal condensation, we performed 2 independent isolations of sEVs from approximately 80 trigeminal ganglia dissected at approximately HH17. To demonstrate sEVs enrichment, we first analyzed the samples by nanoparticle tracking analysis (NTA) and TEM. The results show that an enrichment of sEVs with mode size of 84 nm (sEV1) and 199 nm (sEV2) in diameter (Fig 7A) and the typical cup-shape of exosomes in the samples (Fig 7B).

Next, we isolated RNAs from these samples to determine if miR-203 is loaded into sEVs. It has recently been demonstrated that miRNAs are specifically loaded into sEVs or retained in the cells by RNA-binding proteins based on the presence of EXOmotifs or CELLmotifs, respectively [41]. To investigate whether this is the case during trigeminal ganglion condensation, we evaluated by RT-PCR the expression of miR-203 (having the EXOmotif GGAC) and miR-34a-5p (having the CELLmotif CAGU) in dissected TGs and isolated sEVs (Fig 7C). The results show that miR-203 is detectable in both trigeminal ganglion samples (TG1 and TG2) as well as in the isolated extracellular vesicles (sEV1 and sEV2). In contrast to miR-203, and in agreement with the presence of the CELLmotif, miR-34a-5p was only detected in the TG and not in sEVs. These results demonstrate that endogenous miR-203 is loaded into sEVs at the time of trigeminal ganglion condensation.

## Discussion

The development of the trigeminal ganglion involves a series of organized events to ensure its correct positioning, aggregation, and differentiation [6], including intimate interactions between NC and placode cells. In our study, we demonstrate that this communication is at least partly mediated by the sEVs released by NC cells that are engulfed by and function within placode cells. Moreover, the selective cargo of miR-203 into sEVs, and possibly other miRNAs, acts as a signaling molecule required for neural crest-placode communication during their aggregation to form the trigeminal ganglion. While this interchange of sEVs between cell types has not previously been noted during development, it has been studied during tumor metastasis [42,43].

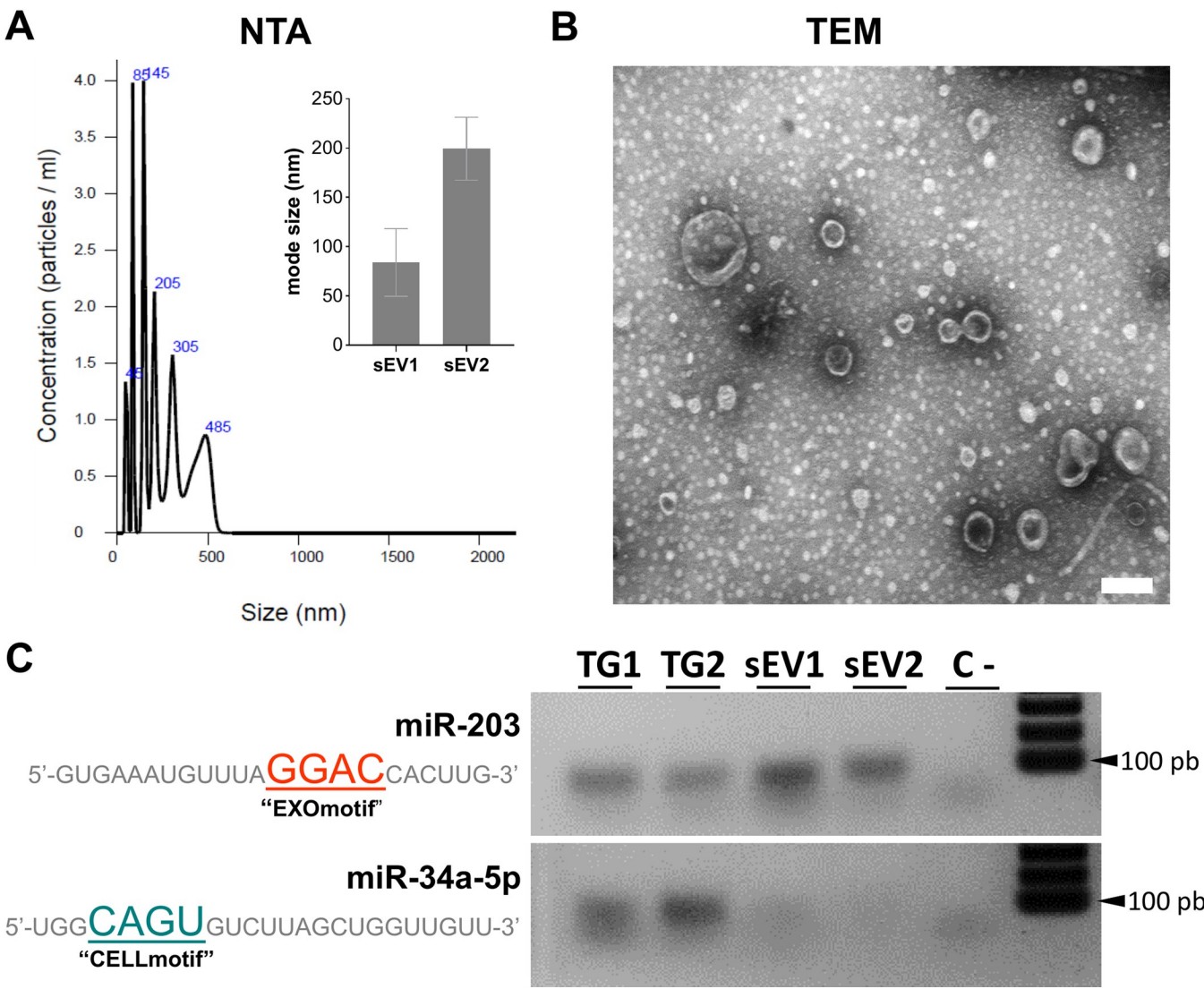

**Fig 7. Endogenous miR-203 is selectively loaded into sEVs produced during TG coalescence. (A)** Graphs show particle concentration, size, and mode size (value are mode ± SD) using NTA from enriched sEVs samples (sEV1 and sEV2) isolated from dissected trigeminal ganglia at HH17 (see S1 Data). **(B)** TEM image showing isolated exosomes after sEVs purified from dissected trigeminal ganglia. Scale bar: 100 nm. **(C)** RT-PCR from 2 independent whole dissected TG (TG1 and TG2) and isolated sEVs from condensing TG (sEV1 and sEV2) raised against the endogenous miR-203 (containing the EXOmotif) and miR-34a-5p (containing the CELLmotif). NTA, nanoparticle tracking analysis; sEV, small extracellular vesicle; TEM, transmission electron microscopy; TG, trigeminal ganglion.

During formation of peripheral ganglia, mesenchymal migratory cells undergo several cellular and molecular changes. Our previous work demonstrated that the epigenetic repression of miR-203 is required for pre-migratory NT cells to initiate an EMT [25]. In this context, miR-203 may reflect a pro-epithelial or pro-aggregation cue. Consistent with this idea, reactivation of miRNAs that are characteristic of the aggregate state, such are miR-203, may be required for proper cell aggregation during organogenesis in early development, as well as during the formation of secondary tumors [24]. The process of trigeminal ganglion aggregation has been likened to a mesenchymal-to-epithelial transition (MET) which is the reverse of EMT. However, placode and NC cells do not become completely epithelial during gangliogenesis, such that this transition may more appropriately reflect a mesenchymal-to-ganglionic

transition [44]. Together, our work has positioned miR-203 expression as a reversible mechanism mediating neural crest EMT during emigration and MET during trigeminal ganglion condensation.

Here, we show that miR-203 functionality is not limited to the neural crest during EMT, but rather plays a later role in regulating condensation with placode cells, thus highlighting the importance of intercellular communication during trigeminal ganglion assembly. In this regard, it is well known that coordination between these cells is critical, and NC cells act as a guide and scaffold for the organization of placode cells to ensure proper assembly [34,45] and the correct orientation of their projections to the central nervous system [45,46].

Neural crest cells communicate with each other and with the extracellular matrix through cell contact, macropinocytosis, filopodia, and via gap junctions [47–50]. Our data combining in vivo and ex vivo approaches reveal an additional mechanism of communication whereby NC cells release EVs that can regulate behavior of placode cells. This is consistent with a recent study demonstrating that cranial NC cells produces EVs that are critical for their migration [51], albeit neither the contents nor the molecular function were addressed. Altogether, sEVs offer a delivery method for cell-to-cell communication in which miRNAs produced and released by donor cells are taken up by recipient cells where they can cause changes in gene expression [13,41,52]. Importantly, a recent study found that the miRNA population released in sEVs is distinct from the population found in the cells of origin [41]. Moreover, the authors demonstrated that miRNAs content enriched in sEVs differs by cell type and identified the presence of EXOmotifs and CELLmotifs, which may be recognized by RNA-binding proteins which participate in the specific sorting mechanism. In our study, we demonstrate that miR-203, presenting a consensus EXOmotif, is sorted into sEVs and can affect the expression of a sensor mRNA expressed in placode cells. This aligns with our findings from experiments utilizing CRISPR/Cas9 and sponge loss of miR-203 function, revealing that neural crest cells serve as the source while placodal cells act as the site of action for proper condensation of the trigeminal ganglia. Consequently, our investigation stands out as one of the pioneering endeavors employing an in vivo model to probe the involvement of miRNA cargo within sEVs in intercellular communication during early developmental stages.

Finally, we demonstrate that neural crest-produced cytoneme-like structures contact placode cells. This structure participates in contact-mediated cell communication during early organogenesis, where cell protrusions are used to deliver signals [53]. Importantly, it was demonstrated that sEVs are transported along cytonemes [54]. Based on this, we speculate that cytonemes-like structures produced by NC cells may enable a high local concentration of sEVs released near placode cells. Active engulfment of sEVs by placode cells ensures that a sufficient load of miRNAs reach the cytoplasm to mediate target inhibition. Altogether, the neural crest-placode interaction during sensory ganglion development offers an excellent in vivo model system in which to examine the mechanism by which selective miRNAs cargo is delivered into sEVs and transported to specific target cells.

## Limitations of the study

It is essential to acknowledge a limitation inherent in our approach. Our inability to track individual electroporated cells hinders our capacity to elucidate their ultimate fate and discern potential non-cell autonomous effects. Notably, our data does not definitively rule out the possibility that the reduction of mir-203 in neural crest cells could influence condensation independently of mir-203-loaded vesicles.

## Supporting information

**S1 Fig.** Stem-loop RT-qPCR confirm sponge-mediated **(A)** or CRISPR/Cas9 **(B)** loss of miR-203 on the treated side compared with the contralateral side of the same group of embryos. Asterisk (*) indicate significant differences by Student's $t$ test. Values are means (A, $n = 4$; B, $n = 3$) ± SD. Heteroduplex mobility assay PCR (HMA PCR) showed multiple heteroduplex bands from miR-203 gRNA-treated embryos (red arrowhead) and a single band obtained in scrambled gRNA-electroporated embryos.
(TIFF)

**S2 Fig. In ovo electroporation to specifically target neural crest or placode cells.** (A) Representation of HH8 stage chick embryos and transverse views of microinjection into the lumen of the NT, position of the electrodes for electroporation, targeted transfection and transfected neural crest cells, but not placodal cells, after migration. (A') Shows transfected GFP neural crest cells co-localizing with HNK1, but not Tuj1, marker. (B) Representation of HH9 stage chick embryos and transverse views of microinjection on top of the ectoderm region, position of the electrodes for electroporation, targeted transfection and transfected placodal cells, but not neural crest after migration. (B') Shows transfected GFP placodal cells co-localizing with Tuj1, but not HNK1, marker.
(TIFF)

**S3 Fig.** **(A)** Transverse section of HH12 embryos sowing migratory cells electroporated with pHluo and immunostained for HNK1 (magenta). **(A')** Zoom of box in A shows all migratory pHluo+ cells co-localized with HNK1 marker. **(B)** Transverse section through the TG showing NC cells electroporated with pHluo and placode cells immunostained for Tuj1 (magenta). Bar graph shows the percentage of Tuj1+ cells that has pHluo puncta incorporated into their cytoplasm (white arrowheads exemplified Tuj1+/pHluo+ cells) per section (3 to 4 sections have been observed) analyzed in 4 different embryos (see S1 Data). Values are means ± SD.
(TIFF)

**S4 Fig. Scheme of ex ovo electroporation at HH5 to test the dual sensor vector specificity.** **(A)** We injected in the right side the miR-203 overexpressing vector (miR-203 OE) together with the dual-sensor vector (Dual sensor) containing 2 copies of complementary sequences to the mature miR-203 in the 3′ UTR of d4EGFP$_N$, but not in the mRFP$_N$. The surviving embryos showed a drastic reduction in the GFP signal of the sensor in the right side, where miR-203 is overexpressed, compared with the left control side where both GFP and RFP are clearly detected. Successfully overexpress a mature miR-203 was evidenced by in situ hybridization using LNA probes. **(B)** Control embryos were injected in the right side the miR-137 overexpressing vector (miR-137 OE) together with the Dual sensor vector containing 2 copies of complementary sequences to the mature miR-203 in the 3′ UTR of d4EGFP$_N$, but not in the mRFP$_N$. The surviving embryos showed in both sides similar EGFP and GFP intensities. Successfully overexpress a mature miR-137 was evidenced by in situ hybridization using LNA probes.
(TIFF)

**S5 Fig.** **(A)** Transverse section of HH17 embryos to identify migratory NC cells with cytoplasmic EGFP+ (black arrowhead), placode cells with nuclear EGFP+/RFP+ (white arrowhead) or only RFP+ (dotted circles), and DAPI staining to visualize the nuclei. **(B)** Time-lapse of co-cultured placode ectodermal cells (electroporated with the dual-colored sensor vector) and dorsal NT (electroporated with miR-203 OE or Control empty vectors) explants. Placodal cells (PC) interact with NC cells (NC) where the EGFP channel was pseudocolored

(red>white>blue) to visualize the EGFP decay in placode cells over time. In control explants PC1-5 maintain high levels of EGFP fluorescence over the time, compared with the decay observe in PC1-6 in miR-203 OE explants.
(TIFF)

**S1 Movie. Time-lapse of co-cultured NC (yellow) with placode (magenta) explants showing the presence of migrasomes, retraction fibers, and cytonemes-like structures.**
(AVI)

**S2 Movie. Z-stack of co-cultured NC (yellow) with placode (magenta) explants showing placode macropinosomes-like structures in their membrane.**
(AVI)

**S3 Movie. NC cells expressing pHluo were surrounded by numerous extracellular pHluo + puncta (possibly EVs deposits) and trails (possibly migrasomes or retraction fibers).** Also, the placode cells showed intra-cytoplasmic vesicular structures containing pHluo + puncta.
(AVI)

**S4 Movie. Z-stack from S3 Movie showing that extracellular vesicles produced by NC cells (pHluo+ in yellow) that are captured and incorporated into the cytoplasm of placode cells (magenta).**
(AVI)

**S5 Movie. Time-lapse about production of filopodia in placode cells that capture and engulf EVs deposits produced by NC cells expressing pHluo+.**
(AVI)

**S6 Movie. 3D reconstruction of placode protrusion demonstrating the internalized pHluo + puncta.**
(AVI)

**S7 Movie. Time-lapse of co-cultured placode cells (electroporated with the dual-colored sensor vector) and neural crest cells (overexpressing miR-203) explants.** Placodal cells interact with NC cells where the EGFP channel was pseudocolored (red>white>blue) to visualize the EGFP decay in placode cells over time.
(AVI)

**S1 Data. Source data.**
(XLSX)

**S1 Raw Images. Row agarose gels.**
(TIFF)

## Acknowledgments

We thank all the authors and members in the Laboratory of Developmental Biology at the INTECH for their contribution and helpful discussions during the course of our study. We thank Agustina Ganuza (Associate Technician, CONICET) for her technical assistance in the lab. We thank Dr. Alissa M. Weaver (Vanderbilt University School of Medicine, Nashville, Tennessee, USA) for the pHluo construct. We are very grateful to the directors and students from "Escuela de Educación Secundaria Agraria de Chascomús" for providing fertilized eggs of excellent quality.

## Author Contributions

**Conceptualization:** Yanel E. Bernardi, Marianne E. Bronner, Natalia de Miguel, Pablo H. Strobl-Mazzulla.

**Data curation:** Pablo H. Strobl-Mazzulla.

**Formal analysis:** Yanel E. Bernardi, Rocío Belén Márquez, Marianne E. Bronner, Natalia de Miguel, Pablo H. Strobl-Mazzulla.

**Funding acquisition:** Marianne E. Bronner, Pablo H. Strobl-Mazzulla.

**Investigation:** Yanel E. Bernardi, Estefania Sanchez-Vasquez, Rocío Belén Márquez, Michael L. Piacentino, Hugo Urrutia, Natalia de Miguel, Pablo H. Strobl-Mazzulla.

**Methodology:** Yanel E. Bernardi, Estefania Sanchez-Vasquez, Rocío Belén Márquez, Izadora Rossi, Karina L. Alcântara Saraiva, Antonio Pereira-Neves, Marcel I. Ramirez, Natalia de Miguel, Pablo H. Strobl-Mazzulla.

**Project administration:** Marianne E. Bronner, Pablo H. Strobl-Mazzulla.

**Resources:** Natalia de Miguel.

**Supervision:** Marianne E. Bronner, Natalia de Miguel.

**Writing – original draft:** Yanel E. Bernardi, Pablo H. Strobl-Mazzulla.

**Writing – review & editing:** Yanel E. Bernardi, Estefania Sanchez-Vasquez, Michael L. Piacentino, Antonio Pereira-Neves, Marcel I. Ramirez, Marianne E. Bronner, Natalia de Miguel, Pablo H. Strobl-Mazzulla.

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
