## [Editor Report · Decision Letter 0]

9 Mar 2023

Dear Dr Strobl-Mazzulla, 

Thank you for submitting your manuscript entitled "Extracellular vesicle-localized miR-203 mediates neural crest-placode communication required for trigeminal ganglia formation" for consideration as a Research Article by PLOS Biology, and apologies for our delay in sending you an initial decision. We had wished to discuss your manuscript with an Academic Editor, with relevant expertise, and had a bit of trouble finding someone who was available over this last week. 

Your manuscript has now been evaluated by the PLOS Biology editorial staff, as well as by an academic editor, and I am writing to let you know that we would like to send your submission out for external peer review.

Once your full submission is complete, your paper will undergo a series of checks in preparation for peer review. After your manuscript has passed the checks it will be sent out for review. To provide the metadata for your submission, please Login to Editorial Manager (https://www.editorialmanager.com/pbiology) within two working days, i.e. by Mar 13 2023 11:59PM.

**IMPORTANT: While we think the study is potentially interesting, our Academic Editor has raised two technical concerns with the study that we think undermine the strength of the conclusions and which would need to be addressed before we can consider your manuscript for publication. I have appended his/her concerns below my signature. If you happen to have data that addresses these concerns, on hand, we ask that you please update the manuscript to include this before review (we can extend the deadline for resubmission, as needed). If you do not have the requested data, we are willing to review the manuscript in its current form, with the understanding that we would require these points to be thoroughly addressed before we can consider your manuscript for publication. As a last option, if you would like to address these concerns with new data, before review, and expect the revision to take more than 1 week, we would ask that you withdraw your manuscript and resubmit the revised study to us, as a new submission, when you are ready. 

Kind regards,

Luke

Lucas Smith, Ph.D.

Associate Editor

PLOS Biology

lsmith@plos.org

COMMENTS FROM THE ACADEMIC EDITOR: 

I think it is an interesting study, with some fascinating data. I do have a few concerns about their interpretation.

One is that they make a claim in Figure 4 that they can label exosomes. However, it is not clear to me if that transgenic labels just exosomes or if it labels the membrane (or other membrane structures). Given the claim, it would be important to show the exosomes are actually labeled with approach, perhaps with a clear isolation from those cells. Without a control ensuring the labeling is exosomes, it is hard to make the claim they make them.

The other concern is Figure 6, which they claim to use a sponge. I know that they cite a previous paper, but the efficiency of the sponge is important to know in each experimental paradigm. I do not see it shown in the paper.

---

## [Decision Letter · Decision Letter 1]

27 Apr 2023

Dear Dr Strobl-Mazzulla,

Thank you for your patience while your manuscript "Extracellular vesicle-localized miR-203 mediates neural crest-placode communication required for trigeminal ganglia formation" was peer-reviewed at PLOS Biology. Your manuscript has been evaluated by the PLOS Biology editors, an Academic Editor with relevant expertise, and by several independent reviewers.

In light of the reviews, which you will find at the end of this email, we would like to invite you to revise the work to thoroughly address the reviewers' reports. As you will see below, the reviewers express interest in the manuscript. However, they also identified important gaps that we think would need to be thoroughly addressed before we can consider your study for publication at PLOS Biology. Based on their specific comments and following discussion with the Academic Editor, it is clear that a substantial amount of work would be required to meet the criteria for publication in PLOS Biology. However, given our and the reviewer interest in your study, we would be open to inviting a comprehensive revision of the study that thoroughly addresses the reviewers' comments.

In the section below, I have included some comments from our Academic Editor which we hope will be helpful in guiding the revision. As we understand that this represents a large amount of work, we have relaxed our standard revision time to allow you 6 months to revise your study. We would also understand if you do not wish to take on this revision and instead prefer to take the manuscript elsewhere. To the end, if you choose not to resubmit to PLOS Biology, we would be willing to explore a transfer of a more modest revision of your manuscript, with these reviews to PLOS Genetics (or another journal of your choice). PLOS Genetics is editorially independent and so we cannot guarantee they would be up for this.

>>>Editorial suggestions for the revision:

While we think it is important to address all the reviewers' comments in a response to reviewers, the following statements stand out as key parts of the revised manuscript and our Academic Editor has provided suggestions for how these might be addressed. It will be broadly important to demonstrate additional data for the requested controls.

--Demonstration that mir-203 is expressed in specific cells in higher resolution. Please add higher magnification and resolution images to visualize specific cell populations.

--Demonstration of the efficiency and specificity of electroporation. Please add additional controls that demonstrate "which cells received the plasmids through electroporation?"

--Figure 6: Controls for this experiment such as electroporation with a dual sensor with scrambled target sequences for miR-203 and a mutated miR-203 that is incapable of binding to its target have not been carried out.

--In experiments using the miR-203 sponge, how was loss of miR-203 function verified? Please add additional controls that demonstrate "the expression of the sponge in the target cells" and the efficiency of that sponge.

--Is there a method to ensure that constructs are electroporated into the correct target cells? For example, in Figure 6 the use of an additional NC marker could verify that cells electroporated with the EGFP labeled construct are indeed neural crest.

--Please address this comment with new data or if possible, with text changes: In figure 7, the notion of selective loading is poorly supported. Further controls are needed. For example, dissected TG samples is a heterogenous mix of cells and presumably the vesicles are a heterogeneous mixture from different cell types. Selective sorting of miRNAs into exosomes is only beginning to be understood and this manuscript is not focused on dissecting this phenomenon, thus figure 7 is trying to tackle a much bigger question in the field.

--While the suggestion to investigate mir-203 targets is important, this is likely beyond the scope of the paper. Further, addressing if EVs are picked up by other cells in the migratory path can be addressed in text.

-- We strongly encourage you to add a limitations paragraph to address some of the questions about mechanisms. It will be important to outline what the paper is exactly showing and how it can lead to important insight on mechanism.

Given the extent of revision that would be needed, we cannot make a decision about publication until we have seen the revised manuscript and your response to the reviewers' comments. Your revised manuscript would need to be seen by the reviewers again, but please note that we would not engage them unless their main concerns have been addressed.

As noted, we expect to receive your revised manuscript within 6 months. Please email us (plosbiology@plos.org) if you have any questions or concerns, or envision needing a (short) extension.

**IMPORTANT - SUBMITTING YOUR REVISION**

*Resubmission Checklist*

*Published Peer Review*

*PLOS Data Policy*

*Blot and Gel Data Policy*

Sincerely,

Luke

Lucas Smith, Ph.D.

Associate Editor

PLOS Biology

lsmith@plos.org

REVIEWS:

Reviewer #1: General comments:

This manuscript describes the role of miR-203 a microRNA expressed in the neural crest cells that functions in the placode cells to influence the development of the trigeminal ganglion in the chick embryo. The authors have shown evidence supporting communication between the NC and placode cells whereby the miR-203 is loaded into sEVs secreted from the NC cells which deliver their cargo to the placode cells. This helps in coalescence and condensation of these cells to give rise to the trigeminal nucleus. This study has provided novel insights and revealed new mechanisms whereby the development of the trigeminal nucleus is regulated. However, there are several errors in the figures as well as inadequate explanation provided for certain observations. The detailed review is provided below. This manuscript may be considered eligible for publication only upon revision that addresses all the points listed below:

Detailed comments:

Fig.1:

1. In Fig. 1B, miR-203 is described as being absent from migrating NC cells but high magnification images are necessary for this to be convincing. 

2. Immunostaining with a marker for placode cells should also be used along with in situ hybridization for miR-203 to show whether miR-203 is expression is restored in the placode cells or the NC cells or both.

3. Scale bar is missing in all figure panels of Figure 1.

Fig. 2:

1. Authors have given reference to their previous paper (Sánchez-Vásquez et al., 2019 ) for the strategy of electroporation with miR-203 gain of function and loss of function construct where pCAG-GFP was used along with the desired construct for electroporation. The authors have not provided any information or data for the visualization of electroporated cells in fig 2A'. Either an electroporation marker or in situ for miR-203 is necessary to determine if miR203 overexpressing cells are the ones that have led to the ectopic condensation.

2. Authors have used HNK1 in Fig.1 and Sox10 in Fig.2 as markers for neural crest cells. The authors should provide a rationale for using two different markers to detect the NC cells.

3. Scale bars are missing in all figure panels.

Fig. 3

1. Fig. 3B shows the effect of the loss of function of miR-203 in placodal cells where endogenous expression of miR-203 is absent according to the information given in the introduction. However, it has not been clearly demonstrated which cells (NC or placodal cells) express miR-203.

2. Authors claim that loss of miR-203 in neural crest cells does not lead to any defect in trigeminal ganglion condensation. However, previous work published by the same group (Sánchez-Vásquez et al., 2019) demonstrated early delamination of neural crest cells. An explanation needs to be provided as to why this early delamination was not observed in this case.

3. Marker for electroporation e.g. GFP expression from a co-electroporated construct or the same construct through IRES-GFP is needed to visualize manipulated cells in all experiments.

4. Scale bar is missing from all the panels in this figure.

Fig. 4:

1. A detailed explanation of the experiment should be provided.

2. In Fig. 4F, to be able to see if neural crest cells electroporated with pHluo-CD63-mScarlet are indeed releasing pHluo-positive sEVs, a marker for neural crest cells like Sox10 along with mScarlet+ and GFP+ fluorescence should be shown in transverse sections. Both mScarlet and pHluo are not clearly overlapping with DAPI staining.

3. For Fig. 4G quantification is required to see how many Tuj1+ placode cells are positive for pHluo fluorescence.

Fig. 5:

1. Detailed information on constructs used in this experiment must be provided.

2. Authors have electroporated Neural crest cells with pHluo and co-cultured them with placode cells. Since pHluo fluorescence can only be seen after the vesicle containing pHluo has been secreted outside. Therefore, it is not clear how authors are visualizing the NC cells that have secreted the vesicles containing pHluo. 

3. In Figure 5D, white arrowheads dictate intra-cytoplasmic vesicular structures containing pHluo+ puncta in placode cells. Since pHluo is pH sensitive hence it should not be fluorescent after being endocytosed by placode cells. An explanation for this needs to be provided.

4. Scale bar is missing from all images of this figure.

Fig. 6:

1. In Fig 6A the schematic showing the miR-203 dual sensor, has target sequences for miR-203 incorrectly labeled as sponge sequences. 

2. DAPI is absent from Fig.6B and B.'

3. Controls for this experiment such as electroporation with a dual sensor with scrambled target sequences for miR-203 and a mutated miR-203 that is incapable of binding to its target have not been carried out.

4. In panel Fig. 6B' the placode cells circled with dashed lines are incorrectly labeled as being GFP+/RFP-, rather they should be GFP-/RFP+. 

5. Image in Fig.6B' does not have a clear distinction between neural crest cells and placode cells as both express GFP. Therefore, a marker for neural crest cells shown along with EGFP would be useful.

6. For analyzing the EGFP/RFP intensity for individual placode cell, EGFP + tuj1 /RFP should be calculated to determine how many placode cells are experiencing miR-203 inhibition.

7. For Fig. 6C, images of control experiments are required for proper comparison and analysis of the time course for GFP degradation.

Fig. 7:

1. Adequate information about NTA is not provided in the materials and methods section.

Reviewer #2: In this manuscript, the authors study the interaction between neural crest and placode cells during trigeminal ganglia formation in early development using chick embryos as a model system. Specifically, the authors investigated communication between neural crest and placode cells through transfer of microRNA miR-203 in small extracellular vesicles (sEVs). The results demonstrate that gain and loss of function of miR-203 in neural crest or placode cells, respectively, affects trigeminal ganglia formation. The authors present evidence for the direct transfer of sEVs from neural crest cells to placode cells and show that sEVs isolated from the trigeminal ganglia contain miR-203. This manuscript suggests a novel mechanism of cell-cell communication between neural crest and placode cells. The following comments/questions may help increase the strength and clarity of the manuscript:

1. Including an insert and a zoomed in image in Figure 1 panels B and C may help readers visualize when miR-203 is present vs. absent. Also, this figure only shows a marker for neural crest cells - could a marker for placode cells also be included?

2. It remains unclear (e.g. Figure 1) in which cells miR-203 is being expressed during trigeminal ganglion formation. Is it neural crest, placodal cells or some other derivative of either of those cells?

3. The authors claim that overexpression of miR-203 generates ectopic aggregation of NC cells in figure 2. This appears to contradict their previous paper from 2019 where they claim overexpression inhibits delamination and accumulation of NC cells? Can the authors

---

## [Decision Letter · Decision Letter 2]

6 Feb 2024

Dear Pablo,

As discussed over email, I am writing to re-open the manuscript file for your PLOS Biology manuscript "Extracellular vesicle-localized miR-203 mediates neural crest-placode communication required for trigeminal ganglia formation", so that you can submit a revised version of your manuscript that addresses the last reviewer requests, as detailed in your recent revision plan. 

At this stage, your manuscript is back under active consideration at our journal; please notify us by email if something comes up and you do not intend to submit a revision so that we may withdraw it.

**IMPORTANT - SUBMITTING YOUR REVISION**

*Re-submission Checklist*

*Published Peer Review*

*PLOS Data Policy*

*Blot and Gel Data Policy*

Sincerely,

Luke

Lucas Smith, Ph.D.

Senior Editor

PLOS Biology

lsmith@plos.org

REVIEWS:

Reviewer #1: The authors have addressed most of the technical concerns raised. They have carried out additional experiments which lend further support to the conclusions drawn.

The revised manuscript may now be considered as suitable for publication.

Reviewer #2: We thank the authors for addressing our concerns by including additional data and analysis, supplementary figures, and resources for our understanding. We feel that this manuscript addresses an interesting mechanism involving miRNA within exosomes that communicate between two distinct cell-types as they converge to generate placode tissue. While the data here support the importance of miR-203 in trigeminal ganglion cells, the central conclusion that NC cells secrete EVs loaded with miR-203 to communicate with placode cells remains poorly supported. The authors do present data that miR-203 overexpression in NC cells represses a miR-203 sensor in placode cells, and that EVs from the trigeminal ganglion contain miR-203; however, they fail to show that miR-203 is not expressed in placode cells which could confound their conclusion. Additionally, the authors include data suggesting that blocking miR-203 in NC cells has no effect on trigeminal ganglion condensation which challenges the conclusion that this miRNA is important for NC cell and placode communication during this developmental period. The authors cite limitations that prevent (1) sorting of placode cells and (2) inhibition of EV formation/secretion (e.g. genetic or pharmacological), both of which would significantly improve the support for the conclusions made in this manuscript. To conclude, the idea that two distinct lineages of cells communicate via exosomes to control tissue/organ formation is provocative, thus our critical focus on this point.

Reviewer #3, David M. Feliciano (note, Reviewer 3 has signed this review): The authors of the manuscript adequately answered all questions and responded to all comments. The manuscript is an excellent piece of science working on an extremely challenging set of questions in developmental biology. The experiments provide compelling evidence that EVs function as an important developmental morphogen.

Reviewer #4: The authors have addressed some points raised by the reviewers. Mostly, they toned down their conclusions or provided more explanation.

The issue of tracing precisely the electroporated cells was raised by several reviewers and remains an issue, in particular since the authors aim to provide evidence for non-cell autonomous effects. I do appreciate electroporation as a strategy for functional experiments in chick and am aware that this is used by many labs. However, it is standard in the field to include a lineage tracer. The authors have included some data to increase confidence in their electroporation strategy and argue that GFP does not survive processing of the embryos. This is easy to resolve by using antibodies for post-staining, again done by many groups in the field. 

My enthusiasm for the manuscript remains reduced because of the lack of mechanistic understanding and cell-specific uptake of EVs.

---

## [Editor Report · Decision Letter 3]

23 May 2024

Dear Pablo,

Thank you for your patience while we considered your revised manuscript "Extracellular vesicle-localized miR-203 mediates neural crest-placode communication required for trigeminal ganglia formation" for publication as a Research Article at PLOS Biology. This revised version of your manuscript has been evaluated by the PLOS Biology editors and by the Academic Editor.

Overall, we are satisfied by the new experimental data provided in support of your conclusions and so we are likely to accept this manuscript for publication. However, before we can accept your study, we have a few editorial requests, which we need you to address, in another revision that we think will not take very long. 

**EDITORIAL REQUESTS: 

1) RESPONSE TO REVIEWERS: While we are satisfied by the new crispr study provided in the manuscript, I would note that in our previous discussion of the revision plan we emphasized the need to address reviewer 4's concerns about the lineage tracer, with additional discussion of limitations, and we asked that in responding to reviewer 2's point, you also add a sentence specifically acknowledging the limitations of the NC and PC KO experiments - such as "the data does not strictly exclude the possibility that mir-203 reduction in neural crest cells could impact condensation independent of mir-203 loaded vesicles". I did not see this discussion added to the manuscript. Please add a labeled subsection to your discussion section, titled 'limitations' (or something similar) that discusses these points. 

2) TITLE: We would like to suggest a small tweak to the title to make it a bit more active. Specifically, we suggest you change the title to: 

"miR-203 secreted in extracellular vesicles mediates the communication between neural crest and placode cells required for trigeminal ganglia formation"

3) FINANCIAL DISCLOSURES: Please update your financial disclosures section in your online system to describe the role of any sponsors or funders in the study design, data collection and analysis, decision to publish, or preparation of the manuscript. If the funders had no role in any of the above, include this sentence at the end of your statement: "The funders had no role in study design, data collection and analysis, decision to publish, or preparation of the manuscript."

4) ABSTRACT: Please note that per journal policy, the model system/species studied should be clearly stated in the abstract of your manuscript. 

5) DATA: You may be aware of the PLOS Data Policy, which requires that all data be made available without restriction: http://journals.plos.org/plosbiology/s/data-availability. For more information, please also see this editorial: http://dx.doi.org/10.1371/journal.pbio.1001797

a. Supplementary files (e.g., excel). Please ensure that all data files are uploaded as 'Supporting Information' and are invariably referred to (in the manuscript, figure legends, and the Description field when uploading your files) using the following format verbatim: S1 Data, S2 Data, etc. Multiple panels of a single or even several figures can be included as multiple sheets in one excel file that is saved using exactly the following convention: S1_Data.xlsx (using an underscore).

b. Deposition in a publicly available repository. Please also provide the accession code or a reviewer link so that we may view your data before publication. 

>>Regardless of the method selected, please ensure that you provide the individual numerical values that underlie the summary data displayed in the following figure panels as they are essential for readers to assess your analysis and to reproduce it:

Fig 3A-D; Fig 6D,E; Fig 6A;

Fig S1A,B; Fig S3;

>>Please also ensure that figure legends in your manuscript include information on where the underlying data can be found, and ensure your supplemental data file/s has a legend.

>>Please ensure that your Data Statement in the submission system accurately describes where your data can be found.

6) BLOT/GEL REPORTING: We require the original, uncropped and minimally adjusted images supporting all blot and gel results reported in an article's figures or Supporting Information files. We will require these files before a manuscript can be accepted so please prepare and upload them now. Please carefully read our guidelines for how to prepare and upload this data: https://journals.plos.org/plosbiology/s/figures#loc-blot-and-gel-reporting-requirements

7) CODE: Per journal policy, if you have generated any custom code during the curse of this investigation, please make it available without restrictions upon publication. Please ensure that the code is sufficiently well documented and reusable, and that your Data Statement in the Editorial Manager submission system accurately describes where your code can be found. 

We expect to receive your revised manuscript within two weeks. 

*Published Peer Review History*

*Press*

Sincerely,

Luke 

Lucas Smith, Ph.D.

Senior Editor

lsmith@plos.org

PLOS Biology

---

## [Editor Report · Decision Letter 4]

7 Jun 2024

Dear Pablo,

Thank you for addressing our editorial requests regarding your PLOS Biology manuscript "miR-203 secreted in extracellular vesicles mediates the communication between neural crest and placode cells required for trigeminal ganglia formation" in your last revision, and over email. We have now had a chance to assess the changes made, and we are largely satisfied by the revision. However, when looking through your manuscript, we did notice a couple of last points which we think will need to be addressed before publication. As these may require changes to the manuscript, I am writing to invite a last short revision so that you can provide the necessary updates. 

**EDITOIRAL REQUESTS: 

1) FIGURE 7A: When assessing your revised manuscript, we noticed a discrepancy with the data related to Fig 7A. Specifically, we see that in the text, this data is described as 'median size of ~80 nm', while the data in the figure presents the mode size. And then I see that the underlying data provided in your S1_data file presents both the mean and mode for these experiments. I have discussed this point with the Academic Editor, and overall, we think that it would be best to present the *mean* size for this experiment, and suggest the text and figure panel be updated accordingly. We also think that the bar graph in figure 7A should present the average of your 2 EV experiments, rather than the results of just one of the experiments (although it would be OK to leave the example concentration trace that is also present in that figure). 

2) Thank you for addressing the last reviewer concerns with additional discussion of the limitations they raised. In our previous letter, we requested that this be included as a subsection that was specifically labelled 'limitations' (or something similar) and I would like to re-iterate this request. We think this would be an important step necessary to address the reviewer comments and that it would not detract from the paper. 

3) Thank you for providing me with your updated S1_data and raw images files over email, containing the raw data and gels underlying your figures. I have uploaded these to our system as part of your manuscript file. Please do double check everything with your submission looks OK and that the files are correct. 

4) As a last point, since your S1_data file now has data related to figure 7A and Fig S3, we ask that you update those figure legends to briefly reference the S1_data file. 

Please attend to the abovementioned editorial requests at your earliest convenience, and ideally, we expect to receive your revised manuscript within two weeks. 

*Published Peer Review History*

*Press*

Sincerely,

Luke

Lucas Smith, Ph.D.

Senior Editor

lsmith@plos.org

PLOS Biology

---

## [Editor Report · Decision Letter 5]

17 Jun 2024

Dear Pablo,

Thank you for the submission of your revised Research Article "miR-203 secreted in extracellular vesicles mediates the communication between neural crest and placode cells required for trigeminal ganglia formation" for publication in PLOS Biology, and thanks also for addressing our last editorial requests in this revision. On behalf of my colleagues and the Academic Editor, Cody J. Smith, I am pleased to say that we can in principle accept your manuscript for publication, provided you address any remaining formatting and reporting issues. These will be detailed in an email you should receive within 2-3 business days from our colleagues in the journal operations team; no action is required from you until then. Please note that we will not be able to formally accept your manuscript and schedule it for publication until you have completed any requested changes.

PRESS

Sincerely, 

Luke

Lucas Smith, Ph.D.

Senior Editor

PLOS Biology

lsmith@plos.org